# Transcending Scaling Laws with 0.1% Extra Compute

**Yi Tay**[†]  **Jason Wei**[†]  **Hyung Won Chung**[†]  **Vinh Q. Tran**  **David R. So**[†]  **Siamak Shakeri**

**Xavier Garcia**  **Huaixiu Steven Zheng**  **Jinfeng Rao**[†]  **Aakanksha Chowdhery**

**Denny Zhou**  **Donald Metzler**  **Slav Petrov**  **Neil Houlsby**

**Quoc V. Le**  **Mostafa Dehghani**

Google

`{vqtran,dehghani}@google.com`

## Abstract

Scaling language models improves performance but comes with significant computational costs. This paper proposes UL2R, a method that substantially improves existing language models and their scaling curves with a relatively tiny amount of extra compute. The key idea is to continue training a state-of-the-art large language model on a few more steps with UL2's mixture-of-denoiser objective. We show that, with almost negligible extra computational costs and no new sources of data, we are able to substantially improve the scaling properties of large language models on downstream metrics. In this paper, we continue training a baseline language model, PaLM, with UL2R, introducing a new set of models at 8B, 62B, and 540B scale which we call U-PaLM. Impressively, at 540B scale, we show an approximately 2x computational savings rate where U-PaLM achieves the same performance as the final PaLM 540B model at around half its computational budget (i.e., saving ∼4.4 million TPUv4 hours). We further show that this improved scaling curve leads to "emergent abilities" on challenging BIG-Bench tasks—for instance, U-PaLM does much better on some tasks or demonstrates better quality at much smaller scale (62B as opposed to 540B). Overall, we show that U-PaLM outperforms PaLM on many few-shot setups, including reasoning tasks with chain-of-thought (e.g., GSM8K), multilingual tasks (MGSM, TydiQA), MMLU and challenging BIG-Bench tasks.

## 1 Introduction

There has been significant interest in scaling of language models [Rae et al., 2021, Chowdhery et al., 2022, Brown et al., 2020]. Scaling has inspired new research across multiple fronts, e.g., scaling laws [Kaplan et al., 2020, Hoffmann et al., 2022, Tay et al., 2022a], emergent abilities [Wei

---

[†]Work completed while at Google.

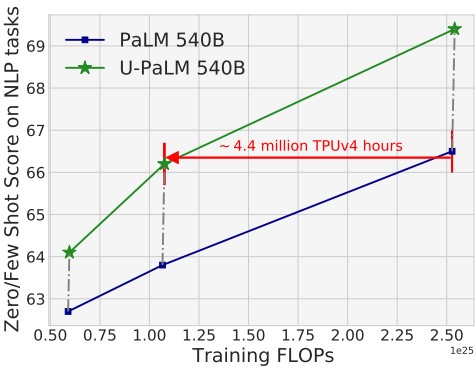

Figure 1: Compute (training flops) versus Quality (average of 20+ NLP zero and few-shot tasks listed in Appendix 11.2). The black dotted line shows the path from initialization from a PaLM checkpoint and training further with UL2R.

et al., 2022a, Ganguli et al., 2022], reasoning capabilities [Wei et al., 2022b, Lewkowycz et al., 2022], inter alia. Generally, scaling laws predict a continued improvement in language model quality as we continue to scale up the computational budget (e.g., bigger models or more data). To date, most large language models that form the basis of scaling law research are trained almost exclusively as left-to-right causal language models [Kaplan et al., 2020, Hoffmann et al., 2022].

This paper proposes a new method to dramatically improve the scaling curves of large language models on downstream performance with a relatively tiny amount of additional computation cost. The key idea is to continue training an existing causal language model [Chowdhery et al., 2022] with a mixture of new objectives—specifically, the UL2 training objective mixture [Tay et al., 2022b]. This restoration is expected to only cost roughly 0.1% to 1% of the original training FLOPs and requires no new data sources, making it highly efficient and convenient. We call this approach UL2R or UL2Restore.

The UL2 objective combines prefix language modeling and long-short span corruption (e.g., infill-

ing) tasks [Raffel et al., 2019] that can be controlled at inference time using a mode switching prompt. Training a large language model with UL2 can be interpreted as teaching it to leverage bidirectional attention (i.e., PrefixLM) or leverage infilling-style pretraining that have been the foundation of language understanding (e.g., T5 [Raffel et al., 2019]). To this end, we postulate that imbuing a state-of-the-art large language model such as PaLM [Chowdhery et al., 2022] with these diverse pretraining schemes as a complement to the original language model objective, enables the model to perform significantly better. Moreover, the UL2 objective enables new prompting capabilities in PaLM which allows it to perform infilling based prompting.

We show that adapting PaLM with UL2R not only results in significantly better scaling laws on well-established few-shot NLP tasks, but also, in our scaling experiments on downstream few-shot tasks, we show that UL2R is two times more efficient (computation savings of approximately 2x) at 540B scale - reaching the performance of the final PaLM 540B model with only half the computation, saving up to 4.4 million TPUv4 hours.

In addition to competitive performance across a range of well-established NLP [Wang et al., 2019], multilingual [Clark et al., 2020a, Shi et al., 2022], and reasoning [Cobbe et al., 2021] benchmarks, we also study the impact of UL2R on a suite of challenging BigBench tasks from Wei et al. [2022a]. Notably, a subset of tasks are described as 'emergent' because PaLM's performance remains flat up to model scale of 62B and only becomes better than non-random at 540B scale. On these set of tasks, we find that UL2R enables (1) doing significantly better at tasks that PaLM struggles at (e.g., navigate, geometric shapes, hyperbaton) and (2) elicits emergent behavior at a smaller scale such as 62B or 8B (e.g., crass ai, vitaminc fact verification). On top of that, U-PaLM strongly outperforms PaLM on some challenging BigBench tasks.

Emergence within the context of large language models is a nascent research area. As the Nobel prize-winning physicist Philip Anderson put it, *'More is different.'* [Anderson, 1972] which describes unpredictable phenomena at different scales. In our context and with mixture-of-denoisers in UL2, we would like to think of this phenomena as *'More is different, but different can also more'* since different pretraining objectives can improve language model quality or elicit new emergent abilities. This work shows that diversity and richer training paradigms can be key to learning new capabilities that were previously hard to acquire with only causal language modeling.

Finally, in addition to emergent task performance and overall improved scaling curves, we show that U-PaLM is also practically more useful since it is equipped with a secondary mode of prompting, i.e., bidirectional infilling. Specifically, UL2R enables a secondary capability for prompting U-PaLM which can be used to fill in more than one blanks in the input prompt. Interestingly, we find that only a small amount of UL2R (e.g., 0.1% tokens or FLOPs) is sufficient to imbue the model with this new capability.

## 2 U-PaLM

This section introduces the technical details of U-PaLM (i.e., PaLM + UL2R). U-PaLM is initialized from PaLM and leverages the same architecture. This section describes the training procedures of UL2R and how they are applied to continue training PaLM. We refer the reader to Section 10 in the Appendix for a comprehensive review of related work.

### 2.1 Training Data

To keep things consistent, we train this model with the same data mixture as PaLM and do not rely on additional sources of data (labeled or unlabeled).

There are three main reasons for this choice. Firstly, we did not want to introduce new tokens to our training process which could conflate findings. Secondly, we did not want to over-index on scaling studies that only measure impact on upstream cross entropy [Hernandez et al., 2022] which claims that repeating data in small quantities could be dis-proportionally harmful. Since the empirical results we obtained are strong, we postulate that repeating tokens could perhaps be not harmful at smaller quantities after all. This is also backed by the continued training of PaLM 62B in [Chowdhery et al., 2022] which showed that repeated data could result in small gains, albeit not as strong as fresh tokens. Thirdly, we consider our data transformation (via UL2) on the training data sufficiently unique and therefore prevents us from explicitly training on the same data with the exact objective or suffering from any memorization issues.

### 2.2 Prefix Language Model Architecture

We train U-PaLM using the prefix language model (PrefixLM) architecture, also sometimes known as

a non-causal decoder-only model. The PrefixLM architecture keeps a non-causal mask in its prefix (or inputs) and applies bidirectional attention to input tokens.

In this architecture, we use a total combined sequence length of 2048 (e.g., PaLM's sequence length) which is then split to 1024 inputs and 1024 targets. In the original UL2 paper and infrastructure, an artifact of its preprocessing pipeline applies padding tokens *first* before combining `inputs` and `targets`. For decoder-only language models, this is inefficient since we would end up with a concatenation of `[prefix]` `[prefix's padding]` `[target]`.

In this work, we optimize the Prefix padding by forcing the model to concatenate prefix and target *before* applying any additional padding. Packing, trimming and padding is then subsequently applied later after the prefix has been concatenated with the targets. Through this *prefix optimization*, we are able to improve example-level sample efficiency of the model.

### 2.3 Loss Objectives

This section describes the setting for the UL2 mixture-of-denoisers that we use in UL2R. The UL2 mixture-of-denoiser objective comprises of three types of denoisers.

- **Regular denoising** whereby the noise is sampled as spans, replaced with sentinel tokens. This is also the standard span corruption task used in Raffel et al. [2019]. Spans are typically uniformly sampled with a mean of 3 and a corruption rate of 15%.

- **Extreme denoising** whereby the noise is increased to relatively *'extreme'* amounts in either a huge percentage of the original text or being very long in nature. Spans are typically uniformly sampled with a mean length of 32 **OR** a corruption rate of up to 50%.

- **Sequential denoising** whereby the noise is always sampled from the start of the text to a randomly sampled point in the text. This is also known as the PrefixLM objective (not to be confused with the architecture).

We kept this simple since many ablations were already explored in Tay et al. [2022b]. We kept the original 7 denoisers as the initial version but later found that a mixture of only three tasks, e.g., 50% PrefixLM, 25% Long (extreme) span corruption, and 25% regular span corruption to be quite simple

and efficient for the setup of continued training. We kept the original mode prompting tokens in the original UL2 design. We used `[S2S]` for S-denoisers (PrefixLM), `[NLU]` for R-denoisers and `[NLG]` for X-denoisers. The 540B U-PaLM model was mainly trained with 50% S-denoiser (PrefixLM), 25% R-denoisers, and 25% X-denoisers.

### 2.4 Training

We train the 540B model for a total of 20k steps with a batch size of 32. We mildly ablate these settings in early experiments with 62B and 8B models but keep them capped within a certain ballpark (e.g., 128 batch size for 50k steps). As a result, this is more similar to *'finetuning'* as compared to full pretraining. The number of additional tokens is therefore very negligible compared to the original pretraining run often coming in at around or less than 0.1% additional compute. The total number of extra tokens we train on for the 540B model is approximately 1.3 billion which constitutes 0.16% extra computation, as the original PaLM model was pretrained on 780B tokens. We use a cosine learning rate decay schedule that anneals the learning rate from $10^{-4}$ to $10^{-6}$. Notably, we also tried a low constant learning rate and found them to perform quite identically. Our U-PaLM 8B and 62B models are trained using 64 TPUv4 chips. Training an U-PaLM 540B model only consumes 512 TPUv4 chips and finishes in about 5 days which is considered to be lightweight.

## 3 Experiments

### 3.1 Improved Scaling Properties on Few-shot Learning

In this experiment, we show improved scaling curves from small amounts of UL2R training on top of both PaLM 8B and PaLM 540B. We use downstream metrics and few-shot evaluation since (1) this is closer to usability of these models and (2) loss with UL2 and causal language modeling is not comparable. We initialized and trained multiple U-PaLM models using different PaLM intermediate checkpoints. On the 8B model, we repeated this 7 times at different intervals. Given that the 540B model was more computationally demanding, we only managed to fit 3 points. For evaluation, we use the average score of NLU and NLG tasks from the GPT-3 suite [Brown et al., 2020]. In total we use 26 tasks (e.g., TriviaQA, NaturalQuestions, Super-GLUE, PIQA, OpenbookQA, ANLI etc). Detailed scores for Figure 2 can be found in the Appendix.

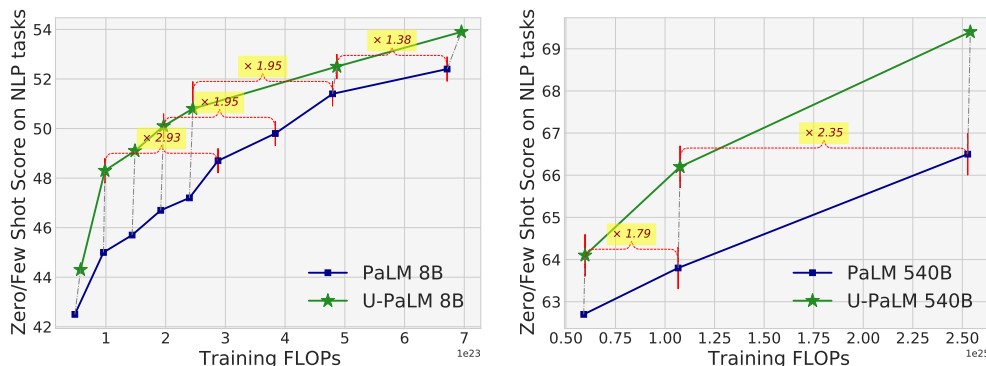

Figure 2: Computation cost (training flops) [Dehghani et al., 2021] versus Quality (average of 20+ NLP zero and few-shot tasks). The dotted line shows the path from initialization from a PaLM checkpoint and training further with UL2R. These plots also present pairs of PaLM and U-PaLM models with comparable/similar performance along with the ratio of PaLM computation cost vs the corresponding U-PaLM computation cost. For example, PaLM 540B trained for $\sim$ 2500 zFLOPs (right most point) took $\sim$ 2.35 times of the computation cost of U-PaLM 540B trained for $\sim$ 1075 zFLOPs, while both models are comparable in terms of performance on zero/few shot on NLP tasks.

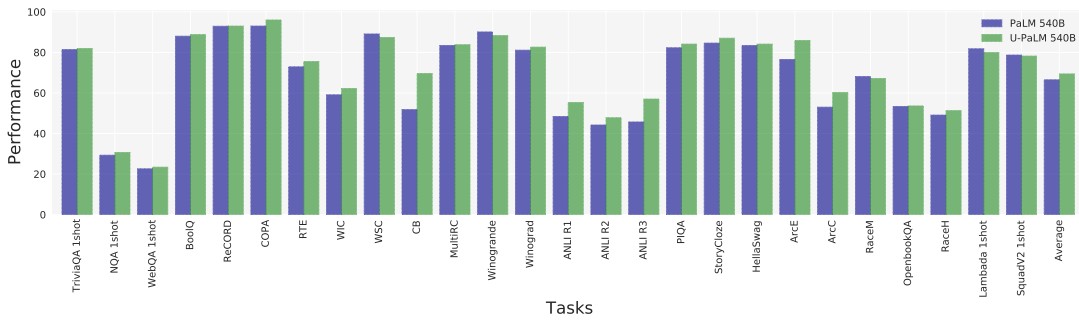

Figure 3: Break down scores of individual zero-shot and one-shot NLP tasks for PaLM and U-PaLM 540B trained for 780B tokens. U-PaLM outperforms PaLM 540B and achieves SOTA on 21 out of 26 tasks.

Figure 2 shows that U-PaLM substantially outperforms the original PaLM models both at 8B scale and 540B scale. Note that the dotted lines represent a pathway before and after UL2R training, we show that UL2R training improves the scaling curve of PaLM substantially, i.e., UL2R provides a more compute-efficient performance improvement compared to training the original PaLM models for longer with the standard causal language modeling objective.

**8B versus 540B** Generally, UL2R consistently improves the underlying PaLM models. Nevertheless, we observe different behaviors on the 8B and 540B models. The gap seems to narrow as the performance of PaLM 8B starts to plateau, i.e., the largest gains are near to the middle of training. As for 540B, the gain continues to grow even at 780B tokens. We believe that this is due to the fact that PaLM 540B still has significant headroom beyond 780B tokens.

**Savings Rate** At a certain stage of training, we have an option to continue training for K more steps using the standard causal language modeling objective OR applying UL2R for a small amount of steps. Here we discuss the counterfactual savings rate of choosing UL2R as opposed to continue training with caussal language modeling. For the 540B model, the saving rates at the middle checkpoint is approximately 2x. This is equivalent to about 4.4 million TPUv4 hours for the 540B model. For the 8B model, the saving rate tend to be lowest at both the start and convergence of the model. It seems to be higher at middle stages of training (relative to convergence) which shows that the utility of UL2R changes with respect to the amount of causal language modeling training already done. For the 540B model, since the PaLM model was not trained to convergence and the number of tokens to parameters ratio is relatively low, the savings rate could still be increasing even beyond 2.35x. Overall, the amount of savings is quite proportionate to the point of training and stage of convergence of the model and can probably be predicted by standard scaling laws [Kaplan et al., 2020, Hoffmann et al., 2022].

Table 1: List of challenging tasks in the BigBench emergent suite (BBES) and corresponding scores of PaLM 540B and U-PaLM 540B. All results are reported with standard 5-shot prompting.

| task | task /reasoning type | PaLM 540B | U-PaLM 540B |
|---|---|---|---|
| navigate | arithmetic, logical | 55.3 | **67.0** (+21.2%) |
| strategyqa | multi-step | 73.9 | **78.3** (+6.0%) |
| crass_ai | commonsense | 97.7 | **100** (+2.4%) |
| logical_sequence | commonsense | **92.3** | 86.5 (-6.7%) |
| vitaminc_fact_verification | contextual, commonsense | 70.2 | **73.9** (+5.3%) |
| understanding_fables | commonsense | 75.7 | **78.4** (+3.6%) |
| identify_odd_metaphor | analogical | 87.2 | **87.5** (+0.3%) |
| hyperbaton | contextual QA | 54.2 | **59.9** (+10.5%) |
| causal_judgment | causal and commonsense | 65.3 | **68.4** (+4.7 %) |
| english_proverbs | commonsense, contextual QA | **91.2** | 87.5 (-4.2%) |
| geometric_shapes | algorithmic, visual | 44.0 | **49.3** (+12.0%) |
| physics_questions | logical, physics, math | 7.6 | **12.5** (+64.5%) |
| snarks | commmonsense | 69.1 | **86.1** (+24.6%) |
| analogical_similarity | analogical | 36.5 | **37.5** (+2.7%) |
| international_phonetic_alphabet_nli | reading comprehension | 65.9 | **68.0** (+3.2%) |
| movie_dialog_same_or_different | commonsense, reading compre. | 64.8 | **68.8** (+6.2%) |
| timedial | commonsense, logical | 78.3 | **81.2** (+3.7%) |
| question_selection | reading comprehension | 54.8 | **59.8** (+9.1%) |
| logical_fallacy_detection | logical reasoning | 80.3 | **81.4** (+1.4%) |
| unit_interpretation | arithmetic, logical | 47.0 | **51.0** (+8.5%) |
| language_identification | multilingual | 36.0 | **38.9** (+8.1%) |
| average (21 tasks) | - | 64.3 | **67.7** (+5.3%) |

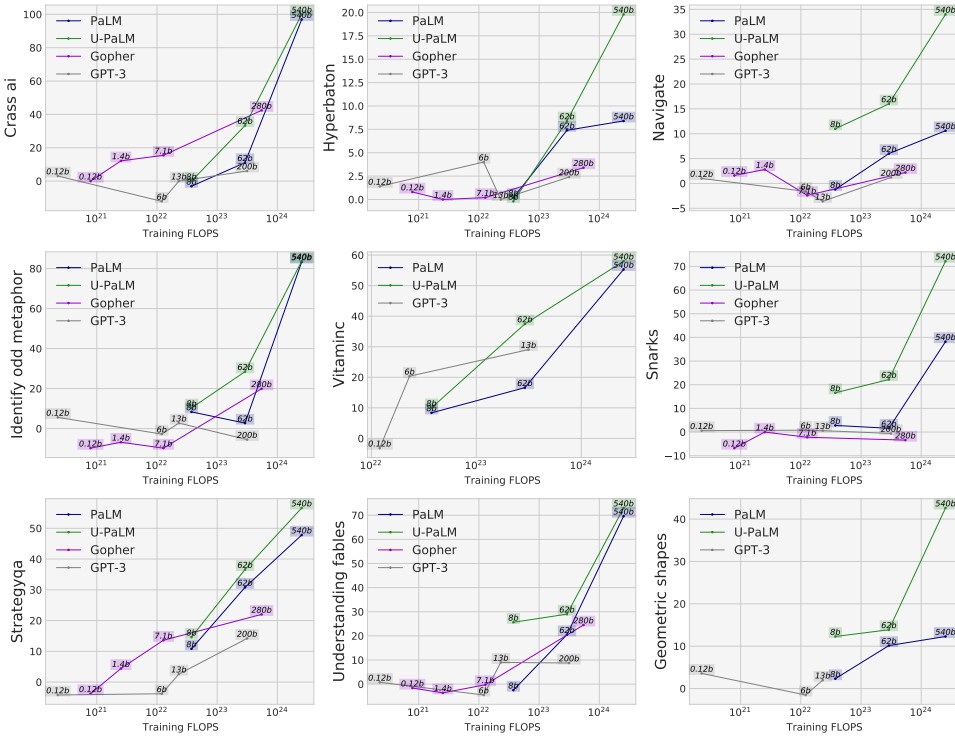

Figure 4: Scaling plots on BIG-Bench emergent suite (BBES) for different sizes of PaLM, U-PaLM, Gopher, and GPT-3 as a function of training FLOPs. Scores are normalized scores where zero denotes more or less random performance. X-axis is in log-scale.

**Breakdown on individual tasks** Figure 3 reports the individual scores on each zero and one-shot task in the mixture. We show that U-PaLM 540B outperforms PaLM 540B on 21 out of 26 tasks. Given that PaLM is the SOTA language model on these tasks, this makes U-PaLM the new state-of-the-art on these tasks.

## 3.2 BigBench Emergent Suite

We select a suite of challenging tasks from BigBench based on a criterion that performance on PaLM on these tasks remain relatively flat-lined at 8B and 62B scale but suddenly unlocks at 540B. We also consider tasks that are difficult for PaLM 540B to solve (near random performance). We call these

Table 2: Results on finetuning on SuperGLUE and TydiQA dev sets.

|  | PaLM 8B | U-PaLM 8B | PaLM 62B | U-PaLM 62B |
|---|---|---|---|---|
| SuperGLUE (Avg) | 83.4 | **86.1** (+3.2%) | 89.5 | **91.4** (+2.1%) |
| TydiQA (EM/F1) | 75.7 / 85.2 | **77.5** (+2.3%) / **86.7** (+1.7%) | 78.3 / 87.3 | **78.4** (+0.1%) / **88.5** (+2.1%) |

Table 3: Results on Massively Multi-Task Language Understanding (MMLU) test set.

| Method | Accuracy |
|---|---|
| Random | 25.0% |
| Average Human Rater | 34.5% |
| GPT-3 5-shot | 43.9% |
| Gopher 5-shot | 60.0% |
| Chinchilla 5-shot | 67.6% |
| PaLM 540B 5shot | 69.3 % |
| U-PaLM 540B 5-shot | **70.7** % (+2.0%) |

suite of tasks EMERGENT suite of BigBench tasks (BBES) as inspired by the criterion set by Wei et al. [2022a]. Note that while these set of tasks overlap but are not entirely identical to BBH [Suzgun et al., 2022]. Moreover, BBES uses the default prompting and templates as BIG-Bench and do not use chain-of-thought prompting. Hence, they are not entirely comparable. BBH results can be found later in section 11.1.3.

### 3.2.1 BIG-Bench Results

Table 1 reports the results of PaLM 540B and U-PaLM 540B on the BigBench emergent suite. We also describe the task and reasoning task for each task. Note that some tasks require a conjunction of various *'skills'* to excel at. For example, the navigate task is a combination of spatial reasoning and arithmetic (counting).

**Overall results and Scaling Plots** We observe that U-PaLM outperforms PaLM on 19 out of the 21 tasks at 540B scale. Moreover, the gains on certain tasks are substantial (e.g., $55.3\% \rightarrow 67.0\%$) on navigate and $69.1\% \rightarrow 86.1\%$ on snarks). On average, there is a $+5.4\%$ relative quality gain on the un-normalized aggregated average across all 21 tasks which we consider to be pretty strong results. Figure 4 which shows the scaling plots of U-PaLM relative to other models. Whenever possible, we also include baselines such as GPT-3 or Gopher from the official BIG-Bench repository.

**UL2R unlocks emergent task performance at smaller scales** Scale (e.g., scaling to 540B) is

known to be one factor that results in emergent task performance [Wei et al., 2022a]. We show that UL2R is able to elicit emergent abilities at smaller scales. For example, the quality on certain tasks such as crass_ai, vitaminc, identify_odd_metaphors are tasks where performance starts to spike at 62B scale (as opposed to only at 540B with the PaLM model. In rarer occasions, the performance of U-PaLM 8B is even higher than PaLM 62B (e.g., snarks, understanding_fables). Overall, these results show that there are strong evidence that *inductive bias* (e.g., combinations of prefix language modeling, span corruption based pretraining in UL2) could be crucial when it comes to unraveling new abilities in large language models.

### 3.2.2 MMLU Results

We compare PaLM and U-PaLM on the Massively Multi-Task Language Understanding (MMLU) benchmark [Hendrycks et al., 2020]. Table 3 reports our results on MMLU's test set. Prior results are reported from [Hoffmann et al., 2022]. Our results show that U-PaLM outperforms PaLM on this task in the 5-shot setup by $2.0\%$ relative gain.

### 3.3 Finetuning

We conduct experiments on SuperGLUE [Wang et al., 2019] and TydiQA [Clark et al., 2020a] finetuning. We conduct experiments at 8B and 62B scale[1]. Fine-tuning is conducted with a constant learning rate for $100k$ steps with a batch size of 32. Table 2 reports finetuning results. We observe that there is substantial improvement in fine-tuning especially at the 8B scale. The gains diminish slightly at 62B scale but are still modest in general. We note that PaLM's fine-tuning performance can be generally considered weaker than expected. For instance, PaLM 8B is generally outperformed by a T5.1.1 large model on the SuperGLUE dev average. We postulate that training PaLM on UL2 and span corruption tasks in complement to causal language

---

[1]Finetuning at 540B is compute intensive and probably less relevant for the finetuning setup since large scale LMs are typically used for prompting. Meanwhile, it is significantly more likely that smaller models are fine-tuned.

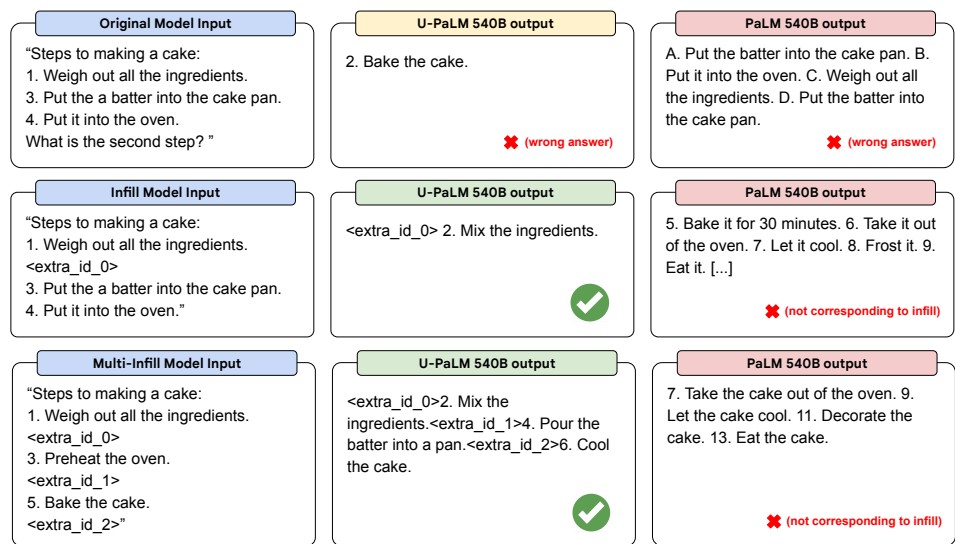

Figure 5: An example of a prompt that is improved by rephrasing to use U-PaLM's infilling capabilities.

modeling can ameliorate some of its flaws. Our results ascertains this by showing that U-PaLM strongly improves quality especially at smaller (8B) scales.

## 3.4 Additional Results & Analysis

We conduct additional, extensive, evaluation and analysis of our approach. Due to space constraints we refer the reader to Section 11 in the Appendix. There we provide results for zero-shot and few-shot NLP tasks including commonsense reasoning, closed book QA & reading comprehension, reasoning & chain-of-thought, and few-shot multilingual tasks. We find that the improvements from U-PaLM over PaLM generally hold across these additional tasks, with major improvements on certain tasks such as GSM8K (+6.6%) [Cobbe et al., 2021], BIG-Bench Hard (+10.7%) [Suzgun et al., 2022], and MGSM (+8.7%) [Shi et al., 2022]. We also include analysis of BBES performance, scaling curves for few-shot experiments, and additional discussion of our methods.

## 4 Qualitative Analysis: New Prompting Capabilities

### 4.1 Infilling Ability

Left-to-right casual language model pretraining has typically allowed models to provide meaningful continuations of prompts. With U-PaLM we observe that, by extending pretraining with a small amount of UL2 denoising steps, the model is also able to pick up infilling abilities – where the model is given a location in the middle of a prompt to fill

in. Notably, with U-PaLM it is possible to query both the infill style and the traditional style via the usage of extra ID tokens (as it is used in denoising) or without, respectively.

In Figure 5, we include example outputs for PaLM, U-PaLM with traditional prompting, as well as U-PaLM with infill prompting. We phrase this particular prompt in two ways: one as a question that is suitable for traditional prompting via PaLM and one leveraging U-PaLM's infill capabilities. In the traditional phrasing, both PaLM and U-PaLM do not produce the correct answer. With the infill phrasing, PaLM ignores the infill token (extra ID token) as PaLM has not seen it during training, and instead produces the rest of the steps after step 4. U-PaLM correctly infills the second step in this example. Finally, a third example is included to demonstrate U-PaLM's ability to infill multiple slots. These examples demonstrate that, with only a small amount of additional training, we are able to expand the functionality of PaLM to serve an entirely new class of queries.

### 4.2 Leveraging Specific Pretraining Modes

Recall that via the UL2 objective, R-, X-, and S-denoisers are associated with the [NLU], [NLG], and [S2S] mode tokens respectively. S-denoisers are essentially the PrefixLM objective, while R- and X-denoisers are variations of span corruption, and thus are also associated with extra ID tokens which we can use during prompting for infill (as shown above.) Given this unique setup, we can control the mode token during inference to gain access to

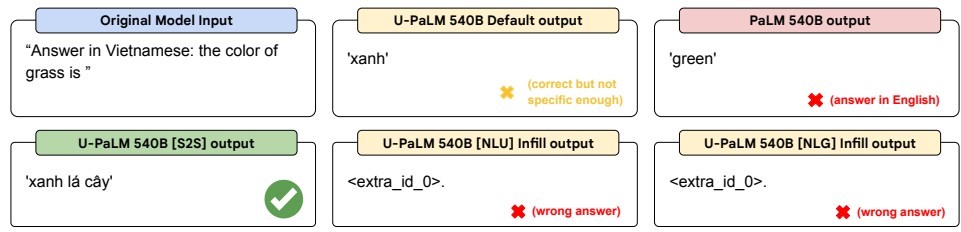

Figure 6: An example of a prompt that works only when querying a specific pretraining mode.

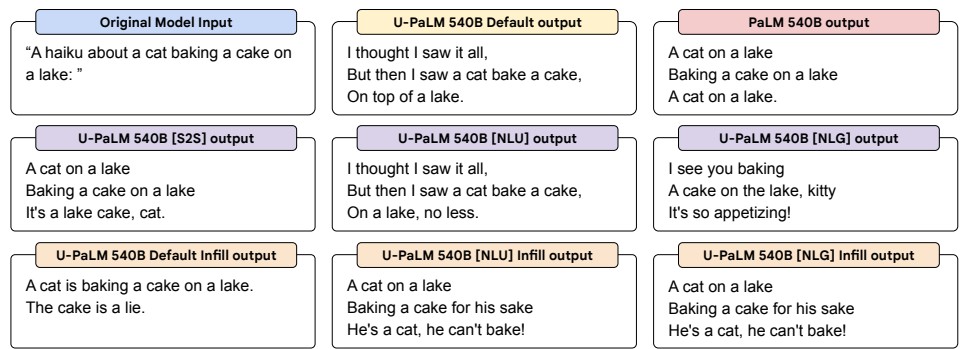

Figure 7: Querying U-PaLM for diverse outputs by using different prompt mode token and LM/infill combinations.

specific knowledge that might have been acquired in one mode but not another. This effectively provides us with more options in how to answer prompts, without the need to make any changes to the learned model or its inference algorithm.

In Figure 6, we include a challenging example where we ask the model to do zero-shot cross-lingual question answering from an English question into a Vietnamese answer. For PaLM and U-PaLM default, we pass the input as-is to the model. For the rest, we prepend one of [S2S], [NLU], or [NLG] to the beginning of the input, and in the case of [NLU] and [NLG], we add the infill token at the end of the input, as typical for these modes. Interestingly, U-PaLM in [S2S] mode is the only variant that returns the correct answer in Vietnamese. Regular PaLM produces the correct answer, but ignores the Vietnamese request, while U-PaLM with default prompting (no mode, no infill) produces a roughly correct answer but could be more specific ('xanh' encompasses both greens and blues). This example shows how accessing specific mode tokens may work well for some prompts more so than others, giving us a powerful technique to serve a larger variety of prompts.

Even though [NLU] and [NLG] modes typically coincide during pretraining with span corruption (involving extra ID tokens, infilling), we can still use [NLU] and [NLG] mode tokens with no infilling at all. Similarly we can use infilling but with

no mode tokens. The variety of ways to prompt U-PaLM results in a useful technique to increase the diversity of the outputs we can get from the model, without resorting to alternative decoding techniques (e.g. sampling). This is particularly useful for more open-ended prompts.

In Figure 7, we ask PaLM and all variants of querying U-PaLM to write a haiku about "a cat baking a cake on a lake" - a very random prompt that the model is unlikely to see during training, yet requires very structured output. All outputs use greedy decoding here, and surprisingly all models generate reasonable haikus about the topic, although not all follow a strict 5-7-5 syllable structure. PaLM's haiku repeats the first and last line, which is somewhat less interesting. We can see that the different combinations of querying U-PaLM results in pleasantly varying poems.

### 4.3 Improved Diversity for Open-ended Generation

Beyond improving the scaling behavior of PaLM, we find that the small amount of continued training applied in UL2R is sufficient to imbue PaLM with new prompting abilities introduced by the UL2 objective. Namely, the use of denoising in UL2 allows PaLM to acquire infilling abilities. Infilling allows U-PaLM to have a second approach to tackling prompts, which we observe to be very useful. In addition, with U-PaLM we can also

supply mode tokens to gain access to specific pretraining objectives. This gives us a powerful tool to control the model without making any updates to the model or its inference. In this section we provide some examples of situations where U-PaLM's expanded prompting capabilities prove to be useful.

## 5 Conclusion

We proposed UL2R for continued training of PaLM. We show that with only $\approx 0.1\%$ additional FLOPs (or compute), we are able to improve the scaling curve and properties of PaLM on many downstream tasks and metrics. Notably, UL2R enables a 4.4 million TPUv4 savings at 540B scale. The resulting model which we call U-PaLM outperforms PaLM on English NLP tasks (e.g., commonsense reasoning and closed-book question answering), reasoning tasks with chain-of-thought, multilingual reasoning, MMLU and a suite of challenging BIG-Bench tasks.

## 6 Limitations

In this work we show the effectiveness of continued training of a 540B PaLM model with UL2R over conditional language modeling alone. We only demonstrate this for the PaLM model and pretraining corpus. Our study is only a demonstration of what is possible with an example near state-of-the-art system, and we do not provide results on what would happen if the underlying model and pretraining corpus were to differ from the one studied here. For example, what would happen if we applied ULR2 to a model that was trained to saturation on a corpus already? Would we observe similar improvements? What would happen if we use a weaker underlying model? This paper also only studies models with 8B+ parameters, and does not provide insight on how UL2R would perform on smaller models and compute regions. We leave these investigations for future work, and this work should not be interpreted as a comprehensive study of continued pretraining or model reuse.

## 7 Ethics Statement

As this work continues training PaLM, we defer discussion of ethical considerations with respect to large language models to the original PaLM paper [Chowdhery et al., 2022]. We do note though that this work presents a way of improving large language models without training from scratch, and all the different types of cost (e.g. environmental) that that might entail.

## 8 Acknowledgements

We thank Le Hou and Oliver Bousquet for their advice and feedback on the paper. We thank Barret Zoph and William Fedus for early discussions about this paper. We thank Adam Roberts for feedback on prior work.

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

# 9    Appendix

# 10    Related Work

**Large language models**    Scaling and improving large language models is one of the most impactful research areas in modern artificial intelligence [Chowdhery et al., 2022]. To this end, large language models not only continue to improve as we scale in terms of data or computational budget [Hoffmann et al., 2022, Kaplan et al., 2020] but also acquire new abilities [Wei et al., 2022a]. The impact of large language models has been ubiquitous and pervasive, unlocking breakthroughs across many fields, e.g., reasoning [Wei et al., 2022b, Wang et al., 2022b, Zhou et al., 2022, Drozdov et al., 2022], math [Lewkowycz et al., 2022], dialog [Thoppilan et al., 2022], multimodal applications [Yu et al., 2022], retrieval [Tay et al., 2022c] *inter alia*.

While there have been many paradigms and self-supervision methods proposed to train these models [Devlin et al., 2018, Clark et al., 2020b, Yang et al., 2019, Raffel et al., 2019], to this date most large language models (i.e., more than 100B parameters) are trained as decoder-only casual language models. For example, flagship large language models such as GPT-3 [Brown et al., 2020], Gopher [Rae et al., 2021] and PaLM [Chowdhery et al., 2022] are all trained as causal language models. Meanwhile, bidirectional models (e.g., BERT [Devlin et al., 2018], T5 [Raffel et al., 2019], ST-MoE [Zoph et al., 2022]) have also been very popular as the goto model of choice, especially in smaller computational regimes (e.g., less than 30B parameters and often times in the ranges of hundred of millions of parameters).

**Scaling laws of large language models**    Kaplan et al. [2020] investigated scaling laws of Transformer language models and first showed the scaling laws are predictive of future performance. The authors found that model size (and not shape) correlates strongly with model quality, i.e., upstream cross entropy. Tay et al. [2021] studied the scaling properties of encoder-decoder models and their impact on upstream and downstream finetuning tasks. Generally, Tay et al. [2021] found that upstream perplexity and downstream quality does not always correlate. As a follow up, Tay et al. [2022a] studied the scaling laws of different model architectures and found that inductive bias does significantly impact the scaling behavior of the model. Finally, Hoffmann et al. [2022] proposed compute-optimal mod-

els that popularized the *'chinchilla'* scaling laws - an approach that aims to be predictive of the optimal amount of data given the number of model parameters. In this work, we mainly consider scaling laws over downstream performance largely because this is more reflective of a language model's usability. Since downstream performance is more important than upstream cross entropy, we advocate for future scaling studies to always incorporate downstream evaluation (and metrics) as opposed to only using cross entropy loss.

**Emergent Abilities**  New behaviors that arise due to scaling language models have been increasingly referred to as *emergent abilities* [Steinhardt, 2022, Ganguli et al., 2022, Wei et al., 2022a]. For instance, Wei et al. [2022a] define emergent abilities as "abilities that are not present in smaller models but as present in larger models." For a few-shot prompted task, this would look like a flat scaling curve (random performance) until a certain critical threshold, during which performance increases to substantially above random. This type of phenomena has been observed across dozens of tasks in the BIG-Bench benchmark [Srivastava et al., 2022]. Although such emergent abilities are typically observed as a function of scale, increasing model scale to induce emergent abilities is computationally expensive. In this paper we show how UL2R unlocks emergence without increasing the number of model parameters.

**Continued Training of Language Models**  The paradigm of continue to train (or finetune) a language model on more data or tasks is commonly known as adaptation. A range of prior work has shown that finetuning language models on a collection of NLP tasks can improve downstream performance on a broad range of downstream tasks [Aghajanyan et al., 2021, Aribandi et al., 2022, Wei et al., 2021, Sanh et al., 2022, Ouyang et al., 2022, *inter alia*]. The majority of this prior work, however, requires additional data such as aggregating dozens or hundreds of NLP datasets [Raffel et al., 2019, Aghajanyan et al., 2021, Aribandi et al., 2022], writing additional templates of instructions [Wei et al., 2021, Sanh et al., 2022], or finetuning on human-labeled annotations [Ouyang et al., 2022]. UL2R does not require new data since it simply re-uses the pre-training data, which makes it orthogonal to continued training methods that leverage large collections of NLP datasets. Adapting a pretrained language model with a new

self-supervised objective has been explored. For example, a model trained with a language modeling objective can be adapted by further training with the masked language modeling objective [Wang et al., 2022a]. The other direction is also possible; a model trained with a masked language objective can be adapted with the causal language modeling objective [Wang et al., 2022a, Lester et al., 2021]. UL2R follows a similar idea but uptrains a language model with a set of diverse and new preordaining tasks from mixture-of-denoisers, even after a vast amounts of standard pretraining and demonstrates a very rapid improvement on variety of setups and tasks.

**Unified language learner (UL2)**  The UL2 [Tay et al., 2022b] model is a state-of-the-art model that bridges both generative causal language models and bidirectional language models. UL2 proposes a mixture-of-denoiser objective that mixes prefix (non-causal) language modeling and infilling (span corruption) within the same model and leverages *mode prompts* to switch between modes during downstream tasks. UL2 is architecture agnostic in which the authors argue that the choice of decoder-only versus encoder-decoder models is largely an efficiency trade-off. In [Tay et al., 2022b], the final UL2 model was trained as a 20B encoder-decoder model, which achieves very compelling performance on both finetuning and in-context learning.

## 11    Additional Results & Analysis

### 11.0.1    Analyzing individual task performance on BIG-Bench

This section dives into individual task performance and attempts to understand quality on different types of BIG-Bench tasks.

**Spatial or Visual Reasoning Tasks** The first category of tasks that U-PaLM does extremely well on are tasks that require some form of spatial or visual reasoning (e.g., `navigate` or `geometric_shapes`). In both of these tasks, U-PaLM 8B outperforms PaLM 540B. We postulate that this is due to the prefix language model architecture and additional PrefixLM training that U-PaLM undergoes. To give a better illustration, consider the following examples from these tasks.

- In the `navigate` task, an example is as follows: *'Turn right. Take 1 step. Turn right. Take 6 steps. Turn right. Take 1 step. Turn right. Take 2 steps. Take 4 steps.'* and the task is a binary classification

task that determines if the agent returns to the starting point.

- In the `geometric_shapes` task, the goal is to predict the shape given an SVG path, e.g., given '*M 31,29 L 34,76 L 82,16 L 31,29*' the model should predict *triangle*.

Here, it is worth noting that both tasks can be improved intuitively by having bidirectional attention and being trained using a PrefixLM like objective. This could explain why U-PaLM could outperform PaLM 540B even at 8B because it was given the right inductive bias.

**Commonsense and Knowledge Tasks** A reasonable portion out of the 21 tasks require some form of commonsense or language-based knowledge in order to do well. It is worth noting that U-PaLM does not train on any new unique tokens (or new data) and therefore, has no access to no new *'knowledge'* compared to vanilla PaLM. Hence, gains here are expected to be milder compared to tasks that rely more on algorithmic or other types of reasoning. However, we observe some relatively smaller gains in certain tasks (e.g., `understanding_fables` or `movie_dialog_same_or_different`). Amongst the tasks in this category, one exception is the `snarks` task which involves detecting sarcasm in natural language. It is worth noting that the only 2 out of 21 tasks where U-PaLM under-performs PaLM belongs to this category (e.g., `logical_sequence` and `english_proverbs`). We think this is reasonable since we do not completely expect UL2R to *always* improve upon this category of tasks given that it does not actually process new data tokens.

**Context Reasoning or Reading Comprehension Tasks** Some tasks require some understanding of context and then requires the language model to answer questions based on this context. An example of this is the `vitaminc_fact_verficiation` task which tries to determine the veracity of a claim given external evidence (context). Another example is the `understanding_fables` task where the goal is to determine the *'morale of the story'* given context (passage or story). It is worth noting that U-PaLM exhibits emergence at 62B scale on these two tasks even though the final 540B model performance is relatively similar. We postulate that this is due to the architectural (and pretraining) advantage of PrefixLM which aids the model in performing much better even at smaller scales. Intuitively, being able

| Task / Model Size FLOPS (ZFLOPS) | PaLM 62B 295.7 | U-PaLM 62B 298.7 | Chinchilla 70B 588 | Gopher 280B 504 | PaLM 540B 2527.2 | U-PaLM 540B 2529.7 |
|---|---|---|---|---|---|---|
| BoolQ 0-shot | 84.8 | 85.4 | 83.7 | 81.8 | 88.0 | **88.8** (+0.9%) |
| PIQA 0-shot | 80.5 | 81.4 | 81.8 | 81.8 | 82.3 | **84.1** (+2.2%) |
| HellaSwag 0-shot | 79.7 | 79.7 | 80.8 | 79.7 | 83.4 | **84.1** (+0.8%) |
| Winogrande 0-shot | 77.0 | 76.2 | 74.9 | 70.1 | 81.1 | **82.6** (+1.8%) |
| Avg. Commonsense | 80.5 | 80.7 | 80.3 | 78.2 | 83.7 | **84.9** (+1.4%) |

Table 4: Results on zero-shot commonsense reasoning.

to bidirectionally reason with context (prefix) could be important in context reasoning tasks.

**Multi-step Reasoning, Analogical Reasoning and Arithmetic tasks** We observe that there are some performance improvements on analogical reasoning task (e.g., `analogical_similarity`) or multi-step reasoning tasks (`strategyqa`) at 540B scale. However, unlike context reasoning tasks, the performance on these class of tasks tend to follow similar scaling patterns albeit with slightly better performance. For example, based on Figure 4, we note that `strategyqa` follows relatively similar scaling curves to PaLM.

## 11.1 Zero-shot and Few-shot NLP

In this section, we evaluate our models on various well-established NLP tasks. These tasks test a spectrum of zero and few-shot abilities of U-PaLM.

### 11.1.1 Commonsense Reasoning

We conduct experiments on four zero-shot commonsense reasoning benchmarks. Specifically, following [Hoffmann et al., 2022], we use BoolQ [Clark et al., 2019], PIQA [Bisk et al., 2020], HellaSWAG [Zellers et al., 2019] and Winogrande [Sakaguchi et al., 2019]. Aside from PaLM 62B and PaLM 540B which we use for direct comparisons with U-PaLM, we also compare with Chinchilla 70B [Hoffmann et al., 2022] and Gopher 280B [Rae et al., 2021]. Table 4 reports the results on zero-shot commonsense reasoning.

We show that U-PaLM 540B outperforms PaLM 540B on all four tasks with an average of (+1.4%) relative improvement and attains the best performance across all models.

### 11.1.2 Question Answering and Reading Comprehension

We evaluate zero-shot and few-shot closed book question answering (CBQA) tasks [Kwiatkowski et al., 2019, Joshi et al., 2017, Roberts et al., 2020] along with the zero-shot Lambada reading comprehension task [Paperno et al., 2016]. Table 5 reports the results of our experiments. We compare with PaLM 62B, PaLM 540B, Chinchilla 70B and

| Task / Model
Size
FLOPS (ZFLOPS) | PaLM
62B
295.7 | U-PaLM
62B
298.7 | Chinchilla
70B
588 | Gopher
280B
504 | PaLM
540B
2527.2 | U-PaLM
540B
2529.7 |
|---|---|---|---|---|---|---|
| TriviaQA 0-shot | 67.3 | 68.3 | 67.0 | 52.8 | **76.9** | 76.4 (-0.7%) |
| TriviaQA few-shot | 72.7 | 73.6 | 73.2 | 63.6 | 81.4 | **82.0** (+0.7%) |
| Natural Questions 0-shot | 18.1 | 18.7 | 16.6 | 10.1 | 21.2 | **21.7** (+2.4%) |
| Natural Questions few-shot | 27.6 | 30.5 | 31.5 | 24.5 | 36.0 | **40.1** (+11.4%) |
| Lambada 0-shot | 75.4 | 79.7 | 77.2 | 74.5 | 77.9 | **80.5** (+3.3%) |
| Avg. QA/RC | 52.2 | 54.3 | 53.0 | 45.1 | 58.7 | **60.1** (+2.3%) |

Table 5: Results on closed book QA and reading comprehension.

| Task / Model | Minerva 540B | PaLM 540B | U-PaLM 540B |
|---|---|---|---|
| GSM8K | 57.8 | 54.9 | **58.5** (+6.6%) |
| BBH | 37.2 | 44.8 | **49.6** (+10.7%) |
| StrategyQA | 61.9 | 76.4 | **76.6** (+0.2%) |
| CSQA | 72.2 | 76.9 | **80.1** (+4.2%) |

Table 6: Experiment results on reasoning and chain-of-thought reasoning experiments.

Gopher 280B. Overall, on few-shot CBQA and reading comprehension, we observe that U-PaLM 540B outperforms PaLM 540B by +2.3% on average and up to +11.4% on few-shot natural questions. Meanwhile, the gain at 62B scale is also strong (i.e., +2.1% on average).

### 11.1.3 Reasoning and Chain-of-thought Experiments

We conduct experiments on reasoning and CoT and compare U-PaLM 540B with PaLM 540B and Minerva 540B. We use the GSM8K [Cobbe et al., 2021], BBH [Suzgun et al., 2022], StrategyQA [Geva et al., 2021] and CommonsenseQA [Talmor et al., 2019] benchmarks. All tasks are run with chain-of-thought (CoT) prompting. Table 6 reports results on reasoning and CoT benchmarks. U-PaLM 540B outperforms both PaLM 540B and Minverva 540B. Notably, the gains on GSM8K and BBH are relatively strong. This shows that U-PaLM does well on reasoning and is well-suited for chain-of-thought reasoning.

### 11.1.4 Multilingual Few-shot Reasoning and Question Answering Tasks

We conduct experiments on few-shot multilingual reasoning and question answering tasks. We use the MGSM (multilingual grade school math) benchmark proposed in [Shi et al., 2022]. For multilingual question answering, we use the well-established TydiQA [Clark et al., 2020a] benchmark. In our experiments, both PaLM 540B and U-PaLM 540B uses chain-of-thought prompting [Wei et al., 2022b]. Table 7 reports our results on MGSM and TydiQA. Our results show that U-PaLM outperform PaLM by a considerable

| Task / Model | PaLM 540B | U-PaLM 540B |
|---|---|---|
| TydiQA | 52.9 | **54.6** (+3.2%) |
| MGSM | 45.9 | **49.9** (+8.7%) |

Table 7: Experiments on Multilingual GSM (MGSM) [Shi et al., 2022] and TydiQA [Clark et al., 2020a]

| Model
Task/#Tokens | PaLM 540B | | | U-PaLM 540B | | |
|---|---|---|---|---|---|---|
| | 182B | 329B | 780B | 182B⁺ | 329B⁺ | 780B⁺ |
| TriviaQA 1shot | 73.4 | 74.4 | 81.4 | 73.3 | 75.6 | **82.0** |
| NQA 1shot | 23.2 | 25.6 | 29.3 | 24.4 | 28.1 | **30.7** |
| WebQA 1shot | 21.6 | 19.9 | 22.6 | 21.0 | 21.7 | **23.4** |
| BoolQ | 82.4 | 85.6 | 88.0 | 85.8 | 88.2 | **88.8** |
| ReCORD | 91.5 | 92.7 | 92.9 | 91.5 | 92.6 | **93.0** |
| COPA | 92.0 | 93.0 | 93.0 | 94.0 | 93.0 | **96.0** |
| RTE | 68.6 | 67.2 | 72.9 | 73.7 | 71.5 | **75.5** |
| WIC | 50.8 | 53.8 | 59.1 | 52.2 | 58.0 | **62.2** |
| WSC | 88.1 | 86.7 | **89.1** | 87.0 | 88.1 | 87.4 |
| CB | 57.1 | 48.2 | 51.8 | 69.6 | 71.4 | **69.6** |
| MultiRC | 76.7 | 81.1 | 83.5 | 78.4 | 81.7 | **83.8** |
| Winogrande | 89.4 | 88.3 | **90.1** | 87.9 | 89.7 | 88.3 |
| Winograd | 76.9 | 79.6 | 81.1 | 78.2 | 79.3 | **82.6** |
| ANLI R1 | 44.3 | 49.4 | 48.4 | 50.3 | 50.6 | **55.3** |
| ANLI R2 | 41.3 | 42.7 | 44.2 | 43.5 | 45.2 | **47.8** |
| ANLI R3 | 43.8 | 42.8 | 45.7 | 46.7 | 49.3 | **57.0** |
| PIQA | 81.0 | 81.9 | 82.3 | 80.8 | 82.0 | **84.1** |
| StoryCloze | 82.7 | 83.9 | 84.6 | 83.7 | 84.2 | **87.0** |
| HellaSwag | 79.1 | 81.8 | 83.4 | 79.5 | 82.3 | **84.1** |
| ArcE | 74.8 | 72.8 | 76.6 | 74.6 | 76.3 | **85.9** |
| ArcC | 48.0 | 46.9 | 53.0 | 48.6 | 50.4 | **60.3** |
| RaceM | 63.6 | 67.3 | **68.1** | 63.2 | 67.1 | 67.2 |
| OpenbookQA | 50.2 | 51.2 | 53.4 | 50.2 | 51.2 | **53.6** |
| RaceH | 45.3 | 48.5 | 49.1 | 45.5 | 48.5 | **51.3** |
| Lambada 1shot | 75.4 | 77.5 | **81.8** | 74.3 | 79.9 | 80.0 |
| SquadV2 1shot | 70.5 | 71.3 | **78.7** | 71.8 | 70.3 | 78.2 |
| Average | 62.7 | 63.8 | 66.5 | 64.1 | 66.2 | 69.4 |

Table 8: Results of PaLM vs U-PaLM at different FLOPs (# tokens) at 540B scale.

margin (+3.2% on TydiQA and +8.7% on MGSM).

### 11.2 Details of Scaling Curves for Few-shot Experiments

We compute a mean aggregated score of the following tasks. We use 21 zero-shot rank classification tasks, i.e., BoolQ, Record, COPA, RTE, WiC, WSC, CB, MultiRC, Winograd, Winogrande, ANLI R1, ANLI R2, ANLI R3, PIQA, StoryCloze, HellaSwag, Arc-E, Arc-C, RaceM, RaceH, OpenbookQA. We use 5 one-shot generative tasks, i.e., TriviaQA, NaturalQuestions, WebQuestions, SQuaDV2 and Lambada. All tasks use the accuracy (or exact match) metric except MultiRC which reports f1a following [Brown et al., 2020]. In total, the aggregated metric is a mean over all **26** tasks. We list the scores that correspond to Figure 2's 540B scaling plot below.

### 11.3 Details of Vocab and Sentinel Tokens

For U-PaLM, we had to train on span corruption or infilling task. We use the same setup as UL2 and T5 where we inject sentinel tokens, e.g., *<extra_id_0>* into the masked positions. In T5, sentinel ids are added as 100 additional vocab tokens at the end of the sentencepiece (vocab). In PaLM, since we restart from an existing PaLM checkpoints, it was quite cumbersome to initialize 100 new embeddings in the vocab. Hence, we opt to simply use the last 100 subwords as sentinel tokens. Finally, we also use eos symbols in the vocab when training the model.

### 11.4 Details of Prompt Templates

As stated in Section 3.2, BBES uses the default prompting and templates a BIG-Bench and do not use chain-of-thought prompting. For full BBH and MMLU results, we use the same set of prompts as [Chung et al., 2022], which we refer the reader to for more details. However, our 5-shot MMLU prompts do not use chain-of-thought, only directly stating the answer option, e.g. "Answer: (C)". Prompts for our zero-shot and few-shot NLP evaluations in Section 10.1 use the same basic templates as [Brown et al., 2020].

### 11.5 Additional Discussion

In this section, we delve into some additional topics and discussions.

#### 11.5.1 What about training from scratch?

We address the elephant in the room. There are multiple perspectives to this question. The first is that UL2R can be thought as a form of *'UL2 schedule'* that sets a single causal language model objective from 0 to $N$ steps and then doing the UL2 mixture from $N$ to $N + \epsilon$. In this sense, if we wanted to train from scratch, this would require modifying the mixture to have significantly more causal language modeling. The second perspective is that UL2R introduces a natural curriculum where the model spents a large fraction of training acquiring basic language modeling before moving on to tasks like infilling or learning how to leverage bidirectional receptive fields. Whether there is a taxonomy or hierarchical of pretraining tasks is still an open question which we hope to answer in future work. The third perspective is simply the practical aspect of U-PaLM. Training a PaLM 540B model from scratch is incredibly costly and we would like to reuse our existing models (or components) as much as possible to design new models for new tasks. U-PaLM is an instance of this type of research. Finally, given that many language models are trained as causal language models, we believe that UL2R presents great opportunity for improving existing models with only a small amount of compute.