# OpenReview forum: "Transcending Scaling Laws with 0.1% Extra Compute"
_EMNLP/2023/Conference — EMNLP 2023 Main_

### Official Review · Reviewer_upoq · 2023-07-20

**Soundness:** 4

**Excitement:**

5: Transformative: This paper is likely to change its subfield or computational linguistics broadly. It should be considered for a best paper award. This paper changes the current understanding of some phenomenon, shows a widely held practice to be erroneous in someway, enables a promising direction of research for a (broad or narrow) topic, or creates an exciting new technique.

**Paper Topic And Main Contributions:**

This paper proposes a method called UL2R to improve the scaling laws and performance of large language models like PaLM.

The key idea is to continue training an existing pretrained LM with a small amount of additional steps using the UL2 training objective.

This provides diversity compared to just training with causal LM, and allows the model to leverage bidirectional attention and infilling-style pretraining.

Experiments show UL2R substantially improves PaLM's scaling curve and performance on NLP tasks, with only minimal extra compute. It also enables emergent abilities at smaller scales and practical benefits like infilling prompting.

**Questions For The Authors:**

A. What would happen if we apply ULR2 to pretraining phrase from scratch?

**Reasons To Accept:**

1. UL2R substantially improves PaLM's performance on downstream NLP tasks.

2. UL2R significantly improves PaLM's scaling. At 540B scale, U-PaLM matches PaLM's performance with only half the FLOPs, saving up to 4.4 million TPU hours.

3. UL2R enables emergent abilities at smaller scales, performing well on certain BigBench tasks at 62B rather than only at 540B.

4. UL2R gives the model new prompting capabilities like bidirectional infilling, improving its flexibility.

**Reasons To Reject:**

1. Only evaluated on a single model (PaLM) so unclear if gains generalize.

2. Limited analysis on why UL2R provides benefits and how different tasks interact

3. Hard to reproduced due to the huge cost.

**Reproducibility:**

2: Would be hard pressed to reproduce the results. The contribution depends on data that are simply not available outside the author's institution or consortium; not enough details are provided.

**Reviewer Confidence:**

4: Quite sure. I tried to check the important points carefully. It's unlikely, though conceivable, that I missed something that should affect my ratings.

---

> ### Author Rebuttal · Authors · 2023-08-28
>
> Thank you for your review and for taking the time to read our paper and provide valuable feedback. We’re glad you enjoyed the paper!
>
> Re: Usage of PaLM. While it is true that we do not evaluate ULR2 on other models, we believe that our selection of PaLM as a baseline model is a strong one with a high chance of findings transferring over to other models as it 1) uses mostly standard mechanisms: a Transformer based decoder-only model pretrained on only causal language modeling over a large text corpus of 780B tokens, 2) spans the largest range of model sizes ever officially disclosed and uses the same recipe for each size, and 3) represents effectiveness of the approach at the very top band of performance, which is more challenging to improve upon.
>
> Re: Analysis of UL2. The design of UL2R was largely influenced by the findings and ablations in the original UL2 paper [1], which we refer to for more detailed analysis of how different tasks interact.
>
> Re: Cost of reproducibility. Although pretraining large models is expensive, applying UL2R is no more expensive than finetuning. For example, for PaLM 8B continued UL2R pretraining happens for 50k steps of 128 batch size. This makes the method relatively cheap to reproduce on other publicly available models.
>
> Re: Using UL2 directly to retrain PaLM from scratch. We discuss this in Section 10.4.1 (Appendix). We leave this to be out of the scope of this paper as it is prohibitively expensive to do so. However, we might refer you to the original UL2 paper which presents detailed results training a 20B encoder-decoder model from scratch with UL2 [1]. Additionally, the PaLM 2 technical report also presents strong results using a UL2-like objective, albeit with less detail [2].
>
> Thanks again for your thoughts and comments.
>
> [1] https://arxiv.org/abs/2205.05131
> [2] https://arxiv.org/abs/2305.10403

---

### Official Review · Reviewer_zMDr · 2023-08-03

**Typos Grammar Style And Presentation Improvements:** 1.The introduction of the article on …
**Soundness:** 5

**Excitement:**

4: Strong: This paper deepens the understanding of some phenomenon or lowers the barriers to an existing research direction.

**Paper Topic And Main Contributions:**

In order to solve the significant computational costs brought by scaling language models, the authors use a slightly improved UL2 to continue training PaLM models from the middle to obtain U-PaLM models. Better results are achieved at a small cost. The authors have tested U-PaLM with different parameters and different stages on multiple tasks, and verified its excellent performance. The article also makes many additional analysis and discussion, and these experiences will enlighten the research in related fields.

**Questions For The Authors:**

1.If using UL2 directly to retrain PaLM from scratch, will it have better results?
2.UL2R compared to PaLM's original training method, where is the extra calculation generated? How is the 0.1% in the article title calculated? This is not clear enough in line 230. And the article seems to be saying that the same results can be achieved with a much smaller cost, so why the emphasis on extra compute?


**Reasons To Accept:**

1.This paper uses PaLM+UL2R to obtain a new U-PaLM model, which surruns PaLM in multiple tasks and achieves SOTA with low extra cost
2.The paper further improves UL2 and proposes prefix optimization
3.The experiment is sufficient and detailed analysis and discussion is made


**Reasons To Reject:**

From the abstract of the article, it seems that the author intends to use UL2R as a general solution to the significant computational costs brought by scaling language models. However, the article only tested it on PaLM, which is not enough to support the universality of the UL2R approach. In fact, this article is more of an introduction to U-PaLM only

**Reproducibility:**

2: Would be hard pressed to reproduce the results. The contribution depends on data that are simply not available outside the author's institution or consortium; not enough details are provided.

**Reviewer Confidence:**

3: Pretty sure, but there's a chance I missed something. Although I have a good feel for this area in general, I did not carefully check the paper's details, e.g., the math, experimental design, or novelty.

---

> ### Author Rebuttal · Authors · 2023-08-28
>
> Thank you for taking the time to read and review our paper! We appreciate the detailed reading and will update the paper to improve the presentation as you mentioned for camera ready.
>
> Re: The usage of PaLM. While it is true that we do not evaluate ULR2 on other models, we believe that our selection of PaLM as a baseline model is a strong one with a high chance of findings transferring over to other models as it 1) uses mostly standard mechanisms: a Transformer based decoder-only model pretrained on only causal language modeling over a large text corpus of 780B tokens, 2) spans the largest range of model sizes ever officially disclosed and uses the same recipe for each size, and 3) represents effectiveness of the approach at the very top band of performance, which is more challenging to improve upon.
>
> Re: Using UL2 directly to retrain PaLM from scratch. We discuss this in Section 10.4.1 (Appendix). We leave this to be out of the scope of this paper as it is prohibitively expensive to do so. However, we might refer you to the original UL2 paper which presents detailed results training a 20B encoder-decoder model from scratch with UL2 [1]. Additionally, the PaLM 2 technical report also presents strong results using a UL2-like objective, albeit with less detail [2].
>
> Re: The 0.1% calculation. PaLM was pretrained on 780B tokens. For PaLM 540B, UL2R trains 20K steps at batch 32, and sequence length 2048, amounting to pretraining for an additional 1.3B tokens, which constitutes 0.16% of the original 780B tokens pretrained. We’ll be sure to update the paper with more details for the final version.
>
> Thank you again for your feedback!
>
> [1] https://arxiv.org/abs/2205.05131
> [2] https://arxiv.org/abs/2305.10403

---

### Official Review · Reviewer_jpZa · 2023-08-11

**Soundness:** 4

**Excitement:**

4: Strong: This paper deepens the understanding of some phenomenon or lowers the barriers to an existing research direction.

**Missing References:**

1. It would help reproducibility if the prompt templates for each dataset was also released (or added to the appendix) this would allow the community to compare directly with this work should anyone would like to asses the approach in their own settings (different model/dataset/multiple model sizes).
2. In section `3.3 Finetuning`, It is mentioned `For instance, PaLM 8B is generally outperformed by a T5 large model on the SuperGLUE dev average. We postulate that training PaLM on UL2 and span corruption tasks in complement to causal language modeling can ameliorate some of its flaws.`. It might be also due to (if it's the original T5 that is mentioned and not T5-v1.1) that SuperGLUE was included in the pretraining process so it could very well be due to that.

**Paper Topic And Main Contributions:**

Instead of continuing to train causal language models with left-to-right, using a collection of multiple training objectives with small amounts of compute compared to the original training compute budget can further boost zero-shot/few-shot model performance on downstream task. The acquired improvements are equivalent to what would otherwise be done by continuing model training with the original objective with substantially more compute, therefore the approach is able to increase performance while also benefitting from massive computational savings.

**Reasons To Accept:**

1. Simple yet effective approach to boost performance at a small compute cost relative to the initial training cost.
2. Allows for a new prompting mechanism to seems to work well.

**Reasons To Reject:**

1. Only uses PaLM which may not be sufficient to claim this approach works in general.

**Reproducibility:**

2: Would be hard pressed to reproduce the results. The contribution depends on data that are simply not available outside the author's institution or consortium; not enough details are provided.

**Reviewer Confidence:**

4: Quite sure. I tried to check the important points carefully. It's unlikely, though conceivable, that I missed something that should affect my ratings.

---

> ### Author Rebuttal · Authors · 2023-08-28
>
> Thank you for reviewing our paper and giving your insightful comments!
>
> Re: The usage of PaLM. While it is true that we do not evaluate ULR2 on other models, we believe that our selection of PaLM as a baseline model is a strong one with a high chance of findings transferring over to other models as it 1) uses mostly standard mechanisms: a Transformer based decoder-only model pretrained on only causal language modeling over a large text corpus of 780B tokens, 2) spans the largest range of model sizes ever officially disclosed and uses the same recipe for each size, and 3) represents effectiveness of the approach at the very top band of performance, which is more challenging to improve upon.
>
> Re: Prompt templates. We are happy to include a subset of the prompt templates in the appendix for camera ready. (For example, for the most important tasks such as MMLU.)
>
> Re: Finetuning comparison against T5. Sorry for the confusion, this comparison is against T5.1.1 which does not use downstream tasks in the pretraining. We will update the text for the final version of the paper.
>
> Thanks again for your feedback!

---

### Official Review · Reviewer_8nct · 2023-08-11

**Soundness:** 5

**Excitement:**

4: Strong: This paper deepens the understanding of some phenomenon or lowers the barriers to an existing research direction.

**Paper Topic And Main Contributions:**

This paper describes a technique for significantly improving the performance of large language models on a variety of downstream NLP tasks by switching training objectives and continuing training for a relatively small amount of time with a mixture of denoisers approach (UL2, thoroughly described in earlier work).

**Reasons To Accept:**

The paper is very well-written, and the experiments seem thorough and convincing. Perhaps most important, this paper has a lot of practical applicability given the cost and energy resources being sunk into training large language models around the world. Further still, the positive results described seem likely to spur a lot of continuation research into similar ways of increasing training efficiency with modified training objectives.

**Reasons To Reject:**

This is quite a strong and thorough paper overall, but the main thing that keeps it from being transformative is a relative lack of novelty: it continues a line of work on large language models and mixture of denoisers, applying the latter for a specific use case (rapidly improving the performance on downstream tasks of language models trained on a strictly left-to-right objective).

**Reproducibility:**

4: Could mostly reproduce the results, but there may be some variation because of sample variance or minor variations in their interpretation of the protocol or method.

**Reviewer Confidence:**

3: Pretty sure, but there's a chance I missed something. Although I have a good feel for this area in general, I did not carefully check the paper's details, e.g., the math, experimental design, or novelty.

---

> ### Author Rebuttal · Authors · 2023-08-28
>
> Thank you for the comments and for taking the time to read and review our paper!
>
> We’re glad that you enjoyed the paper overall. Regarding the novelty of our methods, we might add that the strength of our approach is in its simplicity. Beyond just the small extra compute cost discussed in the paper, the practical cost is also small: one can leverage existing pretraining objectives and dramatically improve their model after-the fact with no architecture changes.

---

### Meta-Review · Area_Chair_GiDD · 2023-09-18

**Recommendation:** 5

**Metareview:**

This paper presents a method called UL2R, which aims to improve the performance of large language models, specifically PaLM, on various downstream NLP tasks. The technique involves continuing the training of a pre-trained language model using a small amount of additional steps with the UL2 training objective. The method claims to achieve significant improvements in performance, scaling, and emergent abilities while providing new prompting capabilities.

Strengths:
* The paper is well-written, with thorough experiments and convincing results.
* UL2R significantly improves PaLM's performance on downstream NLP tasks and scaling, with minimal extra compute.
* The technique enables emergent abilities at smaller scales and provides new prompting capabilities, such as bidirectional infilling.
* The approach offers practical applicability and potential for further research in increasing training efficiency with modified training objectives.

Weaknesses:
* The paper's main limitation is the lack of novelty, as it continues a line of work on large language models and mixture of denoisers.
* The method is only evaluated on a single model (PaLM), which raises questions about the generalizability of the gains.
* Some reviewers found the presentation of innovation points to be vague and repetitive, making it harder to understand the exact contributions.

Overall, this paper is considered strong and potentially transformative, offering valuable insights and practical benefits in improving the performance of large language models. However, further evaluation on other models and a clearer presentation of innovation points would strengthen the claims and generalizability of the method.

---

### Decision · Program_Chairs · 2023-10-07

**Decision:**

Accept-Main

**Comment:**

This paper presents a method called UL2R, which aims to improve the performance of large language models, specifically PaLM, on various downstream NLP tasks. The technique involves continuing the training of a pre-trained language model using a small amount of additional steps with the UL2 training objective. The method claims to achieve significant improvements in performance, scaling, and emergent abilities while providing new prompting capabilities.

Strengths:
* The paper is well-written, with thorough experiments and convincing results.
* UL2R significantly improves PaLM's performance on downstream NLP tasks and scaling, with minimal extra compute.
* The technique enables emergent abilities at smaller scales and provides new prompting capabilities, such as bidirectional infilling.
* The approach offers practical applicability and potential for further research in increasing training efficiency with modified training objectives.

Weaknesses:
* The paper's main limitation is the lack of novelty, as it continues a line of work on large language models and mixture of denoisers.
* The method is only evaluated on a single model (PaLM), which raises questions about the generalizability of the gains.
* Some reviewers found the presentation of innovation points to be vague and repetitive, making it harder to understand the exact contributions.

Overall, this paper is considered strong and potentially transformative, offering valuable insights and practical benefits in improving the performance of large language models. However, further evaluation on other models and a clearer presentation of innovation points would strengthen the claims and generalizability of the method.